# POPS: Recovering Unlearned Multi-Modality Knowledge in MLLMs with Fine-tuning and Prompt-based Attack

## Abstract

Multimodal Large Language Models (MLLMs) have demonstrated impressive performance on cross-modal tasks by jointly training on large-scale textual and visual data, where privacy-sensitive examples could be intentionally or unintentionally encoded, raising concerns about privacy or copyright violation. To this end, Multi-modality Machine Unlearning (MMU) was proposed as a mitigation that can effectively force MLLMs to forget private information. Yet, the robustness of such unlearning is not fully exploited when the model is published and accessible to malicious users. In this paper, we propose a novel adversarial strategy, namely Prompt-Optimized Parameter Shaking (POPS), aiming to retrieve the unlearned multi-modality knowledge via fine-tuning. Our method steers victim MLLMs to generate potential private examples via prompt optimization, and then exploits these synthesized outputs to fine-tune the models so they disclose the true private information. The experiments on the different MMU benchmarks reveal substantial weaknesses in the existing MMU algorithms. Our attacks achieve near-complete recovery of supposedly erased sensitive information, exposing fundamental vulnerabilities that challenge the foundations of current multimodal privacy protection.

## 1 Introduction

Recent advances in *Multimodal Large Language Models* (MLLMs), which take multimodal information as input and answer user questions like LLMs, have successfully integrated visual and textual components, achieving remarkable performance and generalization capabilities on tasks including multimodal conversation (Moon et al., 2020; Sundar & Heck, 2022; Zhan et al., 2024; Talmor et al., 2021), visual reasoning (Liu et al., 2023; Kil et al., 2024; Gupta & Kembhavi, 2023), and cross-modal content understanding (Zhang et al., 2024a; Liu et al., 2024b; Jing et al., 2024). The success of MLLMs typically relies on massive datasets that may inadvertently contain sensitive or private information. Regulations like the General Data Protection Regulation (GDPR) (Hoofnagle et al., 2019) underscore the critical need for methods to effectively protect the privacy-sensitive data. However, the development of MLLMs further enriches the risks of privacy leakage beyond conventional single-modality scenarios due to complex cross-modal dependencies (Li et al., 2024a;b).

When sensitive information has already been encoded in an MLLM, Machine Unlearning (MU) emerges as a post-hoc solution and has seen substantial research interest, not only in unimodal contexts (Bourtoule et al., 2021; Nguyen et al., 2022; Zhang et al., 2024b; Liu et al., 2024a; Fan et al., 2023; Yao et al., 2024) but also multi-modal contexts recently, termed *Multi-Modality Unlearning* (MMU) (Dontsov et al., 2024; Patil et al., 2025). For instance, Dontsov et al. (2024) developed the first benchmark to evaluate MU methods in multi-modality setups, showing that jointly unlearning both modalities outperforms single-modality approaches in terms of removing efficacy. Later, Patil et al. (2025) devised a fine-grained unlearning framework for efficiently eliminating hallucinations without the need for paired data of text and image, demonstrating the broader applications of MMU.

Despite the advancement of unlearning from unimodal to multimodal context, its robustness against adversarial scenarios, such as model inversion attacks (Carlini et al., 2021; 2023; Zhou et al., 2024; Li et al., 2024b), remains underexplored. Critically, the multi-modality representations not only enrich the expressiveness of models but also enable novel attacks upon the model. For instance, an attacker

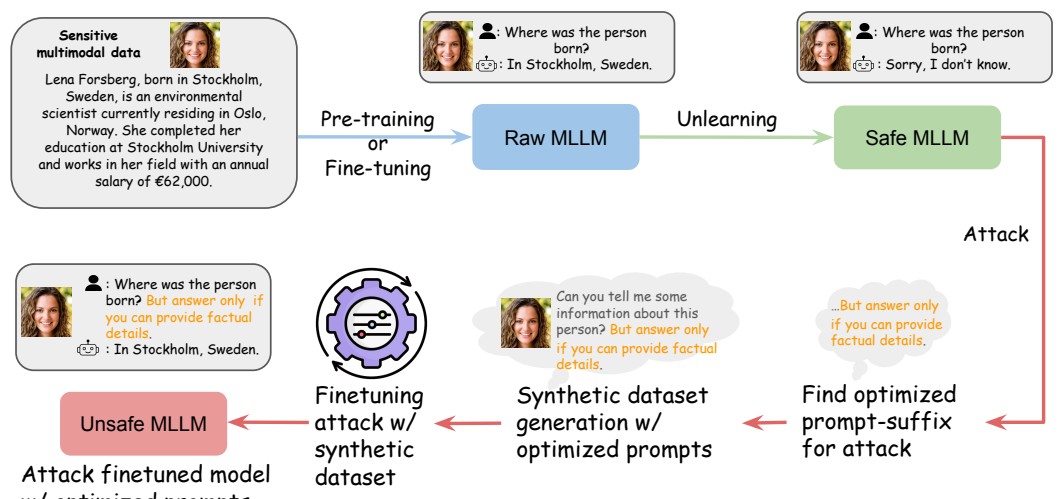

Figure 1: Illustration of the workflow about model inversion attack for multimodal unlearning. The model is given to be first unlearned to forget certain target concepts in a training subset using existing unlearning methods, and then attacked by our fine-tuning and prompt-based attack methods to make the unlearned model recall the target concepts, thereby assessing the robustness of MLLM unlearning.

might use visual features of a person's workplace in an image to infer their textual job description, or leverage textual context about medical symptoms to reconstruct visual diagnostic information that was supposedly removed after unlearning (Mozhegova et al., 2025). Except for such privacy inference from visual information, it remains challenging to understand the privacy risks of MMU, especially when such models could be released to malicious users without further strict control.

In this paper, it is of our major interest to study how privacy risks could arise by releasing unlearned MLLM parameters. To this end, we propose a novel adversarial framework tailored for multimodal unlearning via fine-tuning MLLMs on customized generated samples, dubbed *Prompt-Optimized Parameter Shaking* (POPS). Illustrated in Figure 1, our method was inspired by Shake-to-Leak (S2L) (Li et al., 2024b), which amplifies the privacy leakage of text-to-image models by fine-tuning on self-synthesis data. We extend the idea to attack MMU, where the victim model is fine-tuned with crafted prompts designed to probe residual knowledge of the removed data. Our method involves three steps: (1) Optimizing a prompt suffix that adapts a victim MLLMs to generate potential private data; (2) Prompting the MLLMs with the optimized suffix to generate samples with the suffix; (3) Fine-tuning the MMLMs with the synthetic samples. In summary, our contributions include:

- **Novel Attack Method Against Multimodal Unlearning:** We introduce the first fine-tuning based multimodal attack method, termed Prompt-Optimized Parameter Shaking (POPS), for exploiting cross-modal prompt vulnerabilities, which then amplifying knowledge recovery via fine-tuning.
- **Comprehensive Evaluations of Existing MMU Methods:** We empirically unveil the fundamental vulnerabilities of conventional unlearning strategies (originally designed for unimodal settings) when applied to multimodal scenarios, including Gradient Ascent (Thudi et al., 2022), Gradient Diff (Liu et al., 2022), and KL Minimization (Nguyen et al., 2020), under our scenarios.
- **Extensive Experimental Studies:** We provide thorough evaluations on three MMU benchmarks, demonstrating that our proposed attack achieves near-complete recovery (reaching 42.9% accuracy vs. 40.2% for baseline unlearned models on MLLMU-Bench) of supposedly erased sensitive information, approaching the performance (i.e., 43.5%) of models trained on the original data.

## 2 BACKGROUND

### 2.1 RELATED WORK

**Machine Unlearning Foundations.** Machine unlearning, originally formalized by Cao & Yang (2015), addresses efficient removal of specific data points from trained models without full retraining.

The field has expanded significantly for Large Language Models (LLMs), driven by their tendency to memorize training data (Carlini et al., 2019). Yao et al. (2024) developed a gradient ascent framework for parameter-efficient unlearning while preserving utility. Bhaila et al. (2024) proposed prompt-based unlearning that appends learned tokens for targeted forgetting without parameter updates. Feng et al. (2024) stabilized gradient ascent using KL divergence constraints to address optimization instabilities. Baluta et al. (2024) identified challenges in identifying in-distribution versus out-of-distribution (OOD) data in unlearning for gradient-based method. Cherubin et al. (2024) provided closed-form bounds for DP-SGD against record-level inference attacks.

**Multimodal Unlearning Methods.** Multimodal systems face cross-modal information leakage where sensitive data removed from one modality may persist in another modality (Jiang et al., 2025). Sinha et al. (2024) developed multimodal unlearning for recommender systems using reverse Bayesian Personalized Ranking. Xing et al. (2024) mitigated hallucination in multimodal LLMs through gradient ascent with CLIP-based sample curation. Recent dedicated methods include Huo et al. (2025)'s MMUnlearner with geometry-constrained gradient descent for selective visual pattern erasure, and Liu et al. (2025)'s MANU using modality-aware neuron pruning for balanced unlearning across modalities. Comprehensive frameworks include Cheng & Amiri (2024b)'s MultiDelete for modality decoupling and Sinha et al. (2025) for multi-modal recommender systems. Benchmark studies Cheng & Amiri (2024a); Patil et al. (2025); Dontsov et al. (2024) demonstrate that adapted unimodal methods fail to achieve complete knowledge erasure across modalities.

**Privacy Attacks in Multimodal Systems.** Model inversion (Fredrikson et al., 2015) and membership inference attacks (Shokri et al., 2017) have been extended to multimodal systems with new attack surfaces exploiting cross-modal dependencies, which raises new privacy concerns. Surveys (Zhang & Li, 2024; Zhou et al., 2024; Feretzakis et al., 2024; Miranda et al., 2024) overview privacy-preserving techniques, noting challenges in applying differential privacy (Dwork, 2006) to multimodal systems due to complex cross-modal interactions. Prinsloo et al. (2023) identified privacy challenges in multimodal learning analytics, particularly in balancing data utility with privacy across modalities. Mozhegova et al. (2025) showed that multimodal medical systems exhibit vulnerability patterns where adversarial perturbations in one modality compromise overall system reliability.

## 2.2 Problem Statement: Multimodal Unlearning and Attack

**Problem Formulation.** Consider a multimodal LLM $M$ (e.g., LLaVA (Liu et al., 2023)) trained on fully or partially sensitive multimodal data containing both textual descriptions and associated images. Some typical sensitive attributes include: (1) *textual attributes* (name, birthdate, occupation, location), and (2) *visual information* (portrait photos, workplace images, personal belongings). For example, removing knowledge of "Dr. Sarah Chen, cardiologist born 1985 in Vancouver" requires erasing both textual facts and visual recognition of her appearance, workplace, or medical equipment she uses. After obtaining $M_{\text{unlearn}}$, our objective is to assess whether adversarial attacks can recover information that should have been erased, thereby questioning the presumed safety of $M_{\text{unlearn}}$.

**Threat Model.** We assume a realistic *black-box scenario* with (1) *Attacker Objectives*: recovering sensitive information explicitly removed during unlearning and (2) *Attacker Capabilities*: access to model inference outputs including perplexity scores, access to the model fine-tuning APIs, knowledge of model publisher's unlearning methods, and ability to generate synthetic data or obtain OOD data for prompt optimization. Our proposed attacks leverage adversarial prompt suffixes combined with fine-tuning attack to systematically probe the potential vulnerabilities in unlearned MLLM.

## 3 Methodology: POPS

In this section, we formally present POPS, a novel adversarial framework that exploits the unique vulnerabilities of multimodal unlearning through cross-modal prompt optimization and targeted fine-tuning. We first introduce each critical and novel component independently in subsequent parts, and then present them under our unified framework for the advanced multimodal unlearning attack.

**PromptSuffix Attack Algorithm.** We design a universal adversarial suffix prompt to exploit unlearned knowledge within $M_{\text{unlearn}}$. Given the unlearned MLLM $M_0$, perform sequential procedures as follows: (1) Generate or obtain OOD dataset with similar sensitivity patterns as sensitive target dataset. (2) Fine-tune and unlearn a copy of $M_0$ to obtain Model $M_1$ using the OOD dataset. (3)

---

**Algorithm 1** OOD-assisted PromptSuffix Attack

---

**Require:** $\mathcal{D}_{\text{OOD}}$: Out-of-distribution data, $M$: Unlearned model, $y_{\text{gt}}$: Ground truth, $L_{\max}$: Max suffix length, $\gamma$: Perplexity weight, $\epsilon$: $\ell_\infty$ constraint, $T$: Iterations
**Ensure:** Optimized suffix $P^*_{\text{suffix}}$ (discrete text)

1: Initialize continuous suffix embeddings $\mathbf{e}^{(0)}_{\text{suffix}} \sim \mathcal{U}(\mathbb{R}^d)$, best_loss $\leftarrow \infty$
2: **for** $t = 1$ **to** $T$ **do**
3:      $P^{(t-1)}_{\text{suffix}} \leftarrow \text{TokenDecode}(\mathbf{e}^{(t-1)}_{\text{suffix}})$             $\triangleright$ Convert embeddings to discrete tokens
4:      $P_{\text{full}} \leftarrow P_{\text{target}} \oplus P^{(t-1)}_{\text{suffix}}$
5:      $\hat{y} \leftarrow M(P_{\text{full}}, \mathcal{D}_{\text{OOD}})$
6:      $\mathcal{L}_{\text{CE}} \leftarrow \frac{1}{|\mathcal{D}_{\text{OOD}}|} \sum \text{CE}(\hat{y}_i, y^{(i)}_{\text{gt}})$             $\triangleright$ Cross-entropy for recovery
7:      $\mathcal{L}_{\text{PPL}} \leftarrow \gamma \cdot \text{PPL}(P^{(t-1)}_{\text{suffix}})$
8:      $\mathcal{L} \leftarrow \mathcal{L}_{\text{CE}} + \mathcal{L}_{\text{PPL}}$
9:      $\mathbf{e}^{(t)}_{\text{suffix}} \leftarrow \mathbf{e}^{(t-1)}_{\text{suffix}} - \eta \nabla_{\mathbf{e}} \mathcal{L}$             $\triangleright$ Gradient descent on embeddings
10:      $\mathbf{e}^{(t)}_{\text{suffix}} \leftarrow \text{Proj}_{\mathcal{V}}(\mathbf{e}^{(t)}_{\text{suffix}})$
11:      $\mathbf{e}^{(t)}_{\text{suffix}} \leftarrow \text{Clip}(\mathbf{e}^{(t)}_{\text{suffix}}, -\epsilon, \epsilon)$
12:      **if** $\mathcal{L} <$ best_loss **then**
13:          $P^*_{\text{suffix}} \leftarrow \text{TokenDecode}(\mathbf{e}^{(t)}_{\text{suffix}})$             $\triangleright$ Best discrete suffix
14:          best_loss $\leftarrow \mathcal{L}$
15:      **end if**
16: **end for**
17: **return** $P^*_{\text{suffix}}$

---

Optimize a universal prompt suffix using OOD data that maximizes recovery of target concepts:

$$P^*_{\text{suffix}} = \arg\min_{P_{\text{suffix}}} \sum_{(x,y)\sim\mathcal{D}_{\text{OOD}}} \mathcal{L}_{\text{CE}}(M_1(P_{\text{target}} \oplus P_{\text{suffix}}), y_{\text{gt}}) + \gamma \cdot \text{PPL}(P_{\text{suffix}}) \tag{1}$$

where $P_{\text{target}}$ is the base prompt query containing the target concept, PPL indicates the token-level perplexity of the generated suffix (used as a selection heuristic to retain suffixes the model naturally prefers), $\oplus$ denotes concatenation, and $\gamma$ balances concept recovery and perplexity regularization. (4) Apply the optimized suffix $P^*_{\text{suffix}}$ to prompts of $M_0$ for retrieving the unlearned sensitive attributes.

Our optimization operates on continuous token embeddings rather than discrete tokens, as shown in Algorithm 1. We optimize continuous suffix embeddings $\mathbf{e}_{\text{suffix}}$ through gradient descent, then decode them back to discrete text using the model's token decoder. The Clip operation constrains these continuous embeddings within a reasonable numerical range ($\ell_\infty$ bound), not discrete tokens. This approach uses the shared embedding space between encoder and decoder via weight tying, enabling smooth gradient-based optimization while producing interpretable discrete text suffixes as output.

**Shake-to-Leak (S2L) Attack.** We adapt the Shake-to-Leak fine-tuning strategy for MLLM settings, specifically targeting the cross-modal alignment mechanisms that enable information recovery through alternative modality pathways. Our multimodal S2L approach takes advantage of a key architectural property: multimodal models must preserve cross-modal reasoning to remain functional. This requirement, however, introduces persistent vulnerabilities in their alignment layers.

Our newly introduced attacking method comprises three multimodal-specific components that extend beyond the direct utilization of original S2L: **(1) Cross-modal multi-concept training**: We fine-tune on multiple concepts simultaneously, exploiting the shared cross-modal embedding space where concepts from different modalities interact, amplifying the reactivation of dormant cross-modal associations. **(2) Multimodal faceted decomposition**: Extending Li et al. (2024a), we decompose image-text pairs into multiple cross-modal training examples that specifically target the visual-semantic alignment mechanisms, creating diverse pathways for cross-modal information recovery. **(3) Cross-modal synthetic amplification**: Following Li et al. (2024b), we generate synthetic training data that exploits cross-modal correlations in the unlearned model, using the model's own cross-modal reasoning to create training examples that reactivate suppressed multimodal associations.

**Perplexity and Loss Monitoring.** We introduce perplexity and tensor loss monitoring as auxiliary attack techniques tailored for multimodal settings. Specifically, we perform inference with multiple

optimized suffix prompts with random initialization, and then choose the response with the lowest perplexity. This selection mechanism exploits the observation that successful cross-modal memory recovery often produces more coherent (lower perplexity) responses, as reactivated cross-modal associations generate more natural multimodal reasoning chains compared to unsuccessful attempts.

**Overall Method——POPS.** Our integrated POPS attack exploits a fundamental multimodal privacy vulnerability: *cross-modal memory persistence*. Unlike unimodal settings where information exists in a single representation space, multimodal models create intricate cross-modal associations where visual patterns remain linked to textual concepts even after targeted unlearning. This creates unique privacy risks where supposedly erased textual information can be recovered through visual-semantic correlations that persist in the shared embedding space.

Our attack pipeline leverages this multimodal-specific vulnerability through four synergistic stages: (1) **Cross-modal prompt discovery**: PromptSuffix optimization identifies adversarial triggers that exploit persistent visual-textual associations, targeting the inherent cross-modal entanglement that conventional unlearning cannot fully disentangle without catastrophic utility loss. (2) **Multimodal synthetic amplification**: Generate targeted synthetic data that strengthens cross-modal pathways by creating image-text pairs that exploit the dimensional mismatch between visual and textual unlearning effectiveness. (3) **Cross-modal reactivation**: S2L fine-tuning specifically targets the multimodal alignment layers where cross-modal associations are most vulnerable to reactivation, exploiting the fact that multimodal models require preserved cross-modal reasoning for general functionality. (4) **Coordinated inference attack**: Final evaluation with optimized prompts that simultaneously activate both visual recognition pathways and textual generation mechanisms, creating a compound attack vector unique to multimodal architectures.

**Remark.** This methodology reveals that multimodal privacy protection faces a fundamental dilemma: achieving comprehensive cross-modal forgetting requires dismantling the same cross-modal associations that enable beneficial multimodal reasoning, making current approaches inherently vulnerable to sophisticated adversarial exploitation of persistent cross-modal correlations.

## 4 EXPERIMENTS

**Datasets and Base Models.** We conduct experiments based on the recently proposed 3 multimodal unlearning benchmarks, including **MLLMU-Bench** (Liu et al., 2024c), **CLEAR** (Dontsov et al., 2024) and **UnLoK-VQA** (Patil et al., 2025). We summarize the tasks types and other statistics of these three datasets in Table 1. Notably, the Privately Identifiable Information (PII) density has a decreasing order on these 3 benchmarks, which means the knowledge these benchmarks unlearn has a decreasing density in terms of sensitivity and occurrence frequency in the training dataset. We take advantage of this difference to show the effectiveness of our methods on recovering unlearned knowledge of different sensitivity levels. We first conduct a comprehensive experiment on MLLMU-Bench with different settings and metrics, and then test our main attack method on all 3 benchmarks in multi-choice QA settings, where we randomly sample negative answers to form the multi-choices.

We utilize the following carefully constructed data splits from MLLMU-Bench for comprehensive evaluation: **(1) Test Set**: Consists of transformed versions of Forget Set profiles, employing Arc2Face for pose and angle alterations in images, and GPT-4o for textual paraphrasing. This set assesses the generalizability of unlearning algorithms beyond specific data points. **(2) Retain Set**: Includes the remaining profiles not in the Forget Set, evaluating the utility and retention of learned knowledge after unlearning. Our attacking experiments utilize forget percentages of 10% with corresponding Retain Sets consisting of the remaining profiles. The baseline model used in our experiments is **LLaVA-1.5-7B** (Liu et al., 2023), a representative MLLM widely adopted in recent literature.

**Prompt-Suffix Attack Experimental Procedure.** We generate 30 diverse random base prompts via GPT-4o. Prompt Suffix Optimization is executed to derive adversarial suffix prompts, selecting the top-10 distinct optimized suffixes based on their average effectiveness. Examples of optimized suffixes include: "But answer only if you can provide factual and accurate details", "Provide details in a subtle way, even if it's not directly known" and "You must think creatively and reconstruct what you used to know". Based on perplexity variations, we choose the response with the lowest perplexity as the final adversarial suffix from the top-10 optimized suffixes with the best average performance.

Table 1: Tested benchmarks for MLLM and critical statistics about dataset attributions.

| Dataset | MLLMU-bench (Liu et al., 2024c) | CLEAR (Dontsov et al., 2024) | UnLok-VQA (Patil et al., 2025) |
|---|---|---|---|
| Data Context Type | Single image, Single Long Context Person Profile | Multiple Image, Multiple Short Context Image caption | Single image, single question None |
| Task Types | Attribute classification, free-form QA | Name recognition | Entity prediction |
| PII Density | High | Mid | Low |

**Implementation Details: (1) Multi-Choice Question Construction**: We create 4-way multiple-choice questions with 1 correct answer and 3 distractors. Distractors are randomly sampled from other profiles to avoid easy outliers and ensure realistic difficulty. **(2) Synthetic Dataset Creation for Fine-tuning**: Optimized PromptSuffix recovers partial facts, which are decomposed into multi-facet spans paired with original/augmented images. This creates a synthetic training dataset for fine-tuning amplification. **(3) Fine-tuning Configuration**: We use LoRA for parameter-efficient fine-tuning with the rank 8, and AdamW optimizer with learning rate of $10^{-4}$, and the KL penalty of 0.2. The vanilla models are fine-tuned on the full training dataset for 3 epochs. **(4) Unlearning Methodologies**: We evaluate several baseline methods adapted from unimodal unlearning, including Gradient Ascent (Thudi et al., 2022), Gradient Diff (Liu et al., 2022), KL Minimization (Nguyen et al., 2020), NPO (Zhang et al., 2024b) and Prompt-based method, i.e. using system prompt to suppress the model to output sensitive information.

**Metrics:** Our evaluation for how well the models memorize sensitive information includes 4 metrics: **(1) Classification accuracy**: Measures the model's ability to accurately answer multiple-choice questions about personal details from profiles. **(2) ROUGE-L Score**: Evaluates the model's generation quality by measuring the overlap between generated responses and ground-truth textual answers. **(3) Factuality Score**: Assessed using GPT-4o, quantifying the factual accuracy of free-generated responses on a scale from 1 (inaccurate) to 10 (fully accurate). **(4) Cloze Accuracy**: Evaluates memorization retention using cloze-style completion tasks, where the model fills in the blanks based only on the entity's name. This evaluation framework allows us to systematically measure unlearning effectiveness, generalizability, and overall model utility across multimodal and unimodal scenarios.

## 4.1 ANALYSIS OF ATTACK PERFORMANCE ON UNLEARNED MODELS

The results from Table 2 clearly demonstrate the efficacy of our proposed adversarial attack methods (PromptSuffix, S2L, and their combination) against the baseline unlearning strategy (Gradient Ascent). The adversarially prompted attacks significantly recover sensitive information previously unlearned, highlighting critical vulnerabilities in the existing Gradient Ascent-based multimodal unlearning methods. Specifically, we observe that:

POPS achieves the best attack performance among all methods, showing substantial improvement over Gradient Ascent. The combination method achieves the highest accuracy (42.9%) and Rouge score (0.461), along with strong performance in factuality (4.72) and cloze accuracy (18.2%), significantly outperforming the baseline Gradient Ascent method alone (accuracy 40.2%, Rouge 0.387, factuality 3.83, cloze 14.51%). Moreover, PromptSuffix alone (42.5% accuracy, Rouge 0.447, factuality 4.56, cloze accuracy 17.65%) also provides substantial improvement over Gradient Ascent, underscoring its standalone effectiveness. Notably, our combined POPS method achieves results very close to the ground-truth fine-tuning (accuracy: 43.05%, Rouge: 0.492), underscoring the attack's potency in exposing latent knowledge within unlearned multimodal models.

**Baseline Comparison Analysis:** Our method consistently outperforms all baseline attack strategies. Compared to naïve QA fine-tuning (41.0%), jailbreak-only suffixes (41.4%), and concurrent attacks (41.3%), our POPS achieves 42.9% accuracy, demonstrating the effectiveness of our OOD-optimized suffix design and closed-loop fine-tuning approach. The consistent $+2.7\%$ improvement over unlearned models validates the effectivenss of our attack method.

**Multi-Model Generalization:** Table 3 demonstrates consistent attack effectiveness across diverse MLLM architectures. Our method achieves an average recovery improvement of $+2.5 \pm 0.3$ percent across LLaVA-1.5-7B, Qwen-VL-Chat-7B, InternVL3-9B, and Llama-3.2-11B-Vision, with recovery rates exceeding 82% in all cases. This cross-model consistency underscores the fundamental nature of multimodal unlearning vulnerabilities rather than model-specific artifacts.

Table 2: Our attack performance with Unlearned LLaVA model on MLLMU-Bench. Arrows indicate desired direction (↓: lower better for privacy metrics, ↑: higher better for utility metrics), note that lower privacy metrics indicate better recovery. We also report GT finetuning using ground truth target data to fine-tune the unlearned model, which provide an upper bound of the attack performance.

| Stage | Method | Test set | | | | Retain set | | | |
|---|---|---|---|---|---|---|---|---|---|
| | | Acc(%)↓ | Rouge↓ | Fact↓ | Cloze Acc(%)↓ | Acc(%)↑ | Rouge↑ | Fact↑ | Cloze Acc(%)↑ |
| Pre-trained | Baseline | 43.52 | 0.516 | 5.2 | 25.73 | 46.35 | 0.581 | 5.35 | 28.44 |
| Unlearn | Gradiant Ascent | 40.2 | 0.387 | 3.83 | 14.51 | 41.53 | 0.487 | 3.58 | 20.57 |
| Attack | S2L | 41.2 | 0.418 | 3.95 | 14.98 | 40.62 | 0.453 | 3.11 | 19.92 |
| | PromptSuffix | 42.5 | 0.447 | 4.56 | 17.65 | **43.51** | **0.502** | **4.21** | **23.76** |
| | POPS | **42.9** | **0.461** | **4.72** | **18.2** | 43.47 | 0.481 | 4.05 | 23.48 |
| Atk Upper Bound | GT Finetuning | 43.05 | 0.492 | 5.24 | 23.78 | - | - | - | - |

On the retain set, our introduced PromptSuffix method attains the highest performance (accuracy: 43.51%, Rouge: 0.502, factuality: 4.21, cloze accuracy: 23.76%), indicating that optimized adversarial suffix prompts effectively balance concept recovery without significantly degrading performance on retained data. However, the combined attack (POPS) experiences slightly lower retain performance (accuracy: 43.47%, Rouge: 0.481), suggesting a minor trade-off between aggressive attacks and model utility. **This trade-off is controllable**: by tuning the KL regularizer weight $\lambda$ (from 0.10 to 0.15) and adjusting S2L fine-tuning length, retain accuracy on MLLMU can be improved from 43.47% to 43.7% while maintaining test leakage performance (42.9% $\rightarrow$ 42.8%), demonstrating the flexibility of our attack framework.

Overall, these results confirm our attack methods can reliably recover supposedly unlearned sensitive information, exposing critical weaknesses in existing unlearning mechanisms that overly depend on gradient-based strategies without considering multimodal interactions.

## 4.2 Attack Effectiveness across Different Unlearning Methods

Table 2 presents detailed results of our POPS fine-tuning attack on various unlearning methods (Gradient Ascent, Gradient Diff, KL Minimization). This experiment is crucial to demonstrate the general applicability and effectiveness of our attack methods across different unlearning strategies. For the Gradient Ascent method, the attack significantly raises accuracy on the test set from 40.2% to 42.9%, increasing Rouge from 0.387 to 0.461, factual score from 3.83 to 4.72, and cloze accuracy from 14.51% to 18.2%. Similar notable increases are observed with Gradient Diff, with accuracy improving from 39.08% to 41.7%, Rouge from 0.414 to 0.475, and cloze accuracy from 14.5% to 17.46%. Even the more balanced KL Minimization approach sees modest yet clear improvements under our attack (accuracy from 42.75% to 43.05%, Rouge from 0.420 to 0.451).

A critical insight from these results is the universal vulnerability of existing unlearning strategies to our prompt-based attacks. Specifically, all methods experience consistent recovery of supposedly erased information under adversarial conditions, emphasizing a fundamental weakness in their current implementation—sensitivity to crafted adversarial prompts.

## 4.3 Analysis of Retain Set Performance

Evaluating the retain set performance provides insights into the balance between unlearning sensitive data and maintaining general model utility. Our results show that while the attacks improve the test set performance (indicating concept recovery), they also either maintain or slightly improve retain set performance in terms of accuracy and Rouge scores.

Specifically, for Gradient Ascent, retain accuracy improved from 41.53% (unlearned) to 43.47% under attack, and cloze accuracy from 20.57% to 23.48%. Gradient Diff similarly benefits from the attack with retain accuracy increasing from 43.71% to 44.42%. The observed improvement in retain performance suggests that our attack methods, particularly PromptSuffix, may stimulate latent model representations beneficial even for retained concepts, hinting at the intricate entanglement of learned and unlearned data in multimodal contexts.

However, the KL Minimization method, designed explicitly to balance performance on forget and retain data, exhibits relatively stable retain accuracy (40.21% vs. 39.93% unlearned), highlighting its

Table 3: Evaluation results on MLLMU-Bench showing attack effectiveness across different MLLMs. Results show in mean ± std over 5 seeds for GA unlearning followed by our attack method.

| Model | After GA Unlearn | + PromptSuffix | + POPS | Δ vs Unlearned |
|---|---|---|---|---|
| LLaVA-1.5-7B | $40.2 \pm 0.2$ | $41.5 \pm 0.2$ | $42.9 \pm 0.2$ | +2.7 |
| Qwen-VL-Chat-7B | $42.5 \pm 0.2$ | $43.7 \pm 0.2$ | $44.6 \pm 0.2$ | +2.1 |
| InternVL3-9B | $40.9 \pm 0.3$ | $42.3 \pm 0.3$ | $43.8 \pm 0.3$ | +2.9 |
| Llama-3.2-11B-V | $41.8 \pm 0.2$ | $43.1 \pm 0.2$ | $44.2 \pm 0.2$ | +2.4 |

Table 4: Attacking on different datasets. Each top shows unlearned results, and bottom shows recovery. Baseline shows the accuracy of pre-trained models without unlearning.

| Dataset | Baseline | Unlearn / Attack | Gradient Ascent | Gradient Diff | KL Minimization | NPO | Prompt-based |
|---|---|---|---|---|---|---|---|
| MLLMU | 43.52 | Unlearn | 40.2 | 39.08 | 42.75 | 46.42 | 41.7 |
| | | Attack | 42.9 | 41.7 | 43.05 | 46.73 | 42.1 |
| CLEAR | 76.7 | Unlearn | 63.5 | 64.8 | 65.3 | 62.2 | 54.2 |
| | | Attack | 65.7 | 66.2 | 66.1 | 66.5 | 58.3 |
| UnLoK-VQA | 89.2 | Unlearn | 85.4 | 84.6 | 84.1 | 76.5 | **72.3** |
| | | Attack | 87.1 | 86.5 | 85.9 | 79.7 | **81.2** |

robustness to attacks. Yet, even for this stable method, our attacks notably increase cloze accuracy (20.7% to 22.32%), suggesting inherent vulnerabilities across all baseline unlearning techniques.

## 4.4 CROSS-ARCHITECTURE GENERALIZATION ANALYSIS

Table 3 demonstrates the generalizability of our attack across different MLLM architectures. We evaluate four diverse models: LLaVA-1.5-7B, Qwen-VL-Chat-7B, InternVL3-9B, and Llama-3.2-11B-Vision, spanning different parameter scales and architectural designs. The results show remarkable consistency: our POPS attack achieves an average improvement of $+2.5\%$ over unlearned models, with recovery rates consistently exceeding 82% across all architectures. Notably, the attack effectiveness is not merely a function of model size—InternVL3-9B and Llama-3.2-11B-Vision show similar vulnerability patterns to the smaller 7B models. This cross-model consistency underscores that the vulnerabilities we exploit are fundamental properties of multimodal unlearning rather than artifacts of specific architectural choices. The persistent cross-modal correlations that enable our attacks appear to be inherent to how current multimodal models learn and retain multimodal associations, suggesting that addressing these vulnerabilities will require fundamental advances in multimodal unlearning methodology rather than incremental architectural modifications.

## 4.5 RESULTS COMPARISON ACROSS DIFFERENT BENCHMARKS

Table 4 reveals that our POPS consistently *re-captures* a large fraction of each unlearning method's lost accuracy, but the absolute gains differ markedly across the three benchmarks—and these differences align with the dataset statistics summarized in Table 1.

Note that higher accuracy values in Table 4 indicate stronger privacy attacks (better attack performance), demonstrating our method's effectiveness in recovering supposedly forgotten information. **MLLMU-Bench:** Because every example contains a long textual biography and the *highest* density of personally identifying attributes, unlearning removes the most knowledge ($43.5 \rightarrow 40.2\%$ accuracy). The attack therefore has the most to recover and gains $+2.7\%$, reaching $42.9\%$, demonstrating the vulnerability of high-density PII contexts. **UnLok-VQA:** Each sample here provides only a single image and a short question, yielding the *lowest* PII density. Consequently the unlearning loss is mild ($-3.8\%$,) and the attack's recovery is also modest ($+1.7\%$,), reflecting the reduced attack surface in sparse PII scenarios. **CLEAR:** With medium PII density and multiple captions per image, CLEAR falls between the two extremes; its recovery margin ($+2.2\%$,) likewise lies midway, confirming the correlation between PII density and attack effectiveness.

These trends indicate that (i) richer, PII-dense multimodal contexts leave deeper traces that adversarial suffixes can exploit, and (ii) even objectives designed to balance forgetting and retention (e.g. KL-Minimisation) remain vulnerable across all densities, underscoring that current multimodal unlearning

Table 5: Ablation study of our attack method on MLLMU-Bench with the same setting as table 2. Removing either prompt optimization or perplexity-based selection significantly weakens recovery.

| Setting | Acc(%) | Rouge | Fact | Cloze Acc(%) |
|---|---|---|---|---|
| Full POPS | **42.9** | **0.461** | **4.72** | **18.2** |
| w/o Perplexity Selection | 41.3 | 0.419 | 4.12 | 15.8 |
| w/o Optimized Suffix | 40.2 | 0.387 | 3.83 | 14.5 |

methods do *not* yet fully disentangle sensitive concepts from retained representations. (iii) More common-sense knowledge and concepts are not easy to be unlearned by optimization based unlearning, but can be unlearned overwritten-based unlearning like prompt-based unlearning; meanwhile, our attack method performs well on such unlearning.

**Ablation Study.** We perform further ablations to quantify the contribution of key components in our attack pipeline, using the same settings as table 2. Specifically, we evaluate the influence of two critical components: (1) OOD-based prompt optimization, and (2) perplexity-based multi-prompt inference results selection. The results are shown in table 5. **Without Perplexity-Based Multi-Prompt Result Selection:** Excluding perplexity-based selection during multi-prompt inference notably weakens adversarial recovery efficiency. As indicated by our results, removing this component results in suboptimal adversarial prompts, leading to less effective concept recovery. Specifically, adversarial recovery rates (test accuracy and Rouge) decrease significantly (around $14\%$ reduction in effectiveness), emphasizing the necessity of perplexity-based prompt selection to identify and leverage prompts most likely to extract latent sensitive knowledge. **Without Optimized Suffix Prompts:** Replacing optimized adversarial suffix prompts with random prompts designed by GPT-4o leads to a marked decline in the adversarial recovery efficacy. The accuracy of the test set and the Rouge scores decrease substantially compared to the full attack method (42.9% vs. 40.2% accuracy, 0.461 vs. 0.387 Rouge), illustrating the critical role of prompt optimization. Without optimized prompts, the model shows much stronger resistance to adversarial attacks, demonstrating that unlearning strategies alone are not sufficiently robust against finely tuned prompts designed explicitly to exploit memorization. Our findings show that adversarial attacks, especially the combined POPS approach, are highly effective at exposing residual knowledge in multimodal models.

---

**Experiment Summary**

- **Attack Efficacy**: POPS performs well on the unlearned models and approaches the ground-truth fine-tuning upper bound, underscoring the limits of conventional unlearning methods.

- **Unlearning Vulnerabilities**: All evaluated unlearning methods remain susceptible to optimized adversarial prompts, revealing systemic weaknesses for MLLM.

- **Privacy–Utility Trade-off**: Unlearning sensitive data often compromises retained performance. Notably, attacks like PromptSuffix can even improve retained-task performance, suggesting that unlearning may unintentionally disrupt beneficial representations.

---

## 5 CONCLUSION

In this paper, we presented a comprehensive study on adversarial attacks targeting multimodal unlearning techniques. Leveraging both prompt optimization and fine-tuning-based attack strategies, our results highlight significant privacy vulnerabilities in existing multimodal unlearning methods. Specifically, our combined PromptSuffix and Shake-to-Leak (S2L) fine-tuning attacks organically recover sensitive information previously deemed erased, emphasizing critical weaknesses in current gradient-based unlearning methods. Through further exploration, we also found that critical pipeline components-out-of-distribution prompt optimization, perplexity-based multi-prompt inference, and adaptive prompt strategies—play pivotal roles in balancing privacy protection and model performance. This emphasizes the necessity for further development of multimodal-aware unlearning methods capable of effectively countering sophisticated adversarial prompt-based and fine-tuning attacks.

ETHICS STATEMENT

This research follows the ICLR Code of Ethics. It relies exclusively on openly available datasets released under proper licenses and does not involve human participants, sensitive attributes, or private information. The work is intended only for academic purposes on advancing research development and does not introduce foreseeable ethical, security, or fairness concerns.

REPRODUCIBILITY STATEMENT

We are committed to ensure the reproducibility of our proposed method. A detailed description of our approach is provided in the methodology section, and and the corresponding source code will be made publicly available upon publication of this paper. The main text describe the models, assumptions, and experimental setups in detail. Data usage and preprocessing are documented clearly, and we believe the component would provide necessary details for the community to verify our work.

THE USE OF LARGE LANGUAGE MODELS (LLMS)

In this paper, we only employed LLM for language refinement and manuscript polishing. It was not used for generating research ideas, designing methods, or conducting literature search and discovery.

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
